**Data Availability Statement:** All relevant data are within the paper.

**Funding:** "Support for this study was provided by Pharmacare PLC in the form of salaries for authors YAZ, AK, and MG. The funders had no role in study

# Stability of extemporaneously prepared sitagliptin phosphate solution

**Abdel Naser Zaid** [1]*, **Yara Abu Zaaror**[2], **Aiman Kaddumi**[2], **Mashhour Ghanem**[2], **Nidal Jaradat**[1], **Tharaa Abu Salah**[1], **Sameera Siaj**[1], **Lana Omari**[1]

1 Department of Pharmacy, Faculty of Medicine and Health Sciences, An-Najah National University, Nablus, Palestine, 2 Pharmacare PLC, Ramallah, Palestine

* anzaid@najah.edu

## Abstract

Sitagliptin is a dipeptidyl peptidase-4 (DPP-4) inhibitor that is used orally in conjunction with diet and exercise to control sugar levels in type 2 Diabetes Mellitus patients. This study aimed to extemporaneously prepare SiP solution (1% w/v) using pure Sitagliptin phosphate (SiP) powder and assess its stability according to pharmaceutical regulatory guidelines. Four SiP solutions, coded T1, T2, T3, and T4, were extemporaneously prepared using pure SiP powder as a source of API. The most suitable one, in terms of general organoleptic properties, was selected for further investigations, including stability studies. For this last purpose, samples of the T4 solution were kept under two storage conditions, room temperature (25°C and 60% Relative Humidity) and accelerated stability conditions (40°C and 75% Relative Humidity). Assay, pH, organoleptic properties, related substances, and microbial contamination were evaluated for 4 consecutive weeks. A High-Performance Liquid Chromatography (HPLC) analytical method was developed and validated to be used for the analysis and quantification of SiP in selected solution formulation. The adopted formula had a pH on the average of 3 to 4. During the stability tests, all pH values remained constant. Furthermore, after 4 weeks of storage under both conditions, the SiP concentration was close to 100%. A stable SiP extemporaneous solution was successfully prepared using pure SiP powder. Patients with swallowing problems who use feeding tubes and are unable to take oral solid dosage forms may benefit from this research. Community pharmacists can prepare the solution using sitagliptin powder as the source of the active ingredient.

## Introduction

Diabetes mellitus (DM) is a metabolic disorder characterized by a high blood sugar level resulting from defects in insulin secretion, insulin action, or both [1, 2]. Untreated hyperglycemia is associated with long-term complications including retinopathy with a potential loss of vision, nephropathy leading to renal failure, and peripheral neuropathy with risk of foot ulcers, amputations [3]. Diet and exercise can help some people to manage type 2 diabetes [4]. However, when lifestyle changes aren't enough to lower blood sugar levels, the physician may prescribe suitable antidiabetic drugs. Some of these drugs are administered orally such as α-glucosidase

design, data collection and analysis, decision to publish, or preparation of the manuscript. The specific roles of these authors are articulated in the 'author contributions' section.

**Competing interests:** The authors have read the journal's policy and the authors of this manuscript have the following competing interests: Authors YAZ, AK, and MG are paid employees of Pharmacare PLC. There are no patents, products in development or marketed products to declare. This does not alter our adherence to PLOS ONE policies on sharing data and materials.

inhibitors, Biguanides, DPP-4 inhibitors, Glucagon-like peptides, Sulfonylureas, and Megliti-nides [5]. Gliptins are a group of the most common drugs used for lowering high blood glucose. SiP chemically is 3-amino-1-(3-trifluoromethyl)-6,8-dihydro-5H-(1,2,4) triazolo[4,3-a] pyrazin-7-yl]- 4-(2,4,5-trifluorophenyl) butane-1-phosphoric acid hydrate] (Fig 1) [6].

SiP is the most commonly prescribed gliptin to treat diabetes mellitus, alone or with metformin. SiP binds to the dipeptidyl peptidase 4 enzyme, inhibiting the breakdown of glucagon-like peptide (GLP)-1 and glucose-dependent insulinotropic peptide (GIP) [7]. These hormones are released by the gut after meals and target the pancreas by raising glucose-dependent insulin secretion and decreasing glucagon output from pancreatic α-cells [8]. SiP has many side effects including arthralgia, myopathy, pruritus, pancreatitis and it may increase hepatic enzymes level [9]. Unfortunately, SiP is available in markets only as a solid dosage form which is unsuitable for patients with swallowing problems. In fact, the prevalence of swallowing difficulties is one of the most common problems among pediatric and geriatric patients [10]. This problem will increase in elderly people as a result of many reasons including Alzheimer's disease, carcinoma, and stroke. Patients with this problem will be less compliant and receive less than optimal treatment with tablets and capsules [11]. Patients having swallowing problems or underuse of feeding tubes may face serious challenges when they should be administered solid oral dosage forms. Accordingly, the community or hospital pharmacist should develop or prepare a new convenient oral liquid dosage form for those patients [12–15]

Therefore, the involved pharmacist should raise several questions to manage such cases, such as if the drug is available in a suitable liquid formulation and, if not, whether they can formulate a convenient oral liquid formulation of the drug or not [14, 15]

To the best of our knowledge, only glibenclamide and metformin are available as a liquid dosage form to treat diabetes mellitus. Unfortunately, SiP is only available as a solid dosage

**Fig 1. Chemical structure of SiP.**

form. This study was conducted to prepare a SiP oral liquid dosage form and to evaluate its chemical and physical stability using bulk SiP powder.

## Materials and methods

### Materials

For this study, SiP powder, Sorbitol, Mannitol, peppermint flavor, cherry flavor powder, and strawberry flavor powder were obtained from Pharmacare -PLC Ramallah, Palestine.

### Instruments and chromatographic conditions

The high-performance liquid chromatography (HPLC) system (Dionex, Thermo Scientific) consisted of an ultimate 3000 model pump (SN 8031808), autosampler (SN 8031972), column oven (SN 8031817), PDA detector (SN 8031310), and Chromeleon software V.6.8 (Dionex, Germany). Weights were measured using Oahu's balance (RI097) (Switzerland), and pH was detected by using Mettler-Toledo PH meter (TYPE: MP225) (ID: RI015) (Columbus, US). A multichannel stirrer model (MS -52M) (Jeio-Tech, Korea) was used to confirm dissolving, and a stability chamber for accelerated storage conditions at 40 ˚C and 75% RH (RI 091) (Memmert, Germany) was used to evaluate the quality of the product at these harsh conditions. A UV-spectrophotometer (Jasco V 730, Japan) was used to check the suitable wavelength at which SIP showed the maximum absorbance.

### Chromatographic conditions

To make a mobile phase ratio of 1:1, 500 ml of 0.1 M potassium dihydrogen phosphate buffer (13.6 g potassium dihydrogen phosphate dissolved in 1000 ml of distilled water) and 500 ml of methanol were combined. Phosphoric acid was used to change the pH to 4.5, after which the solution was degassed and filtered through a 0.45 membrane filter. The column oven temperature was 222˚C, the run time was approximately 20 minutes, and the analyte retention time was approximately 6.3 min.

### Formulation of SiP oral solution

A quality by testing (QbT) method was applied to guarantee the desired final quality of the SiP solution. Accordingly, several trials were carried out to achieve the most suitable formula that met the desired quality in terms of compliance of taste and dissolving behavior as shown in Table 1. Initially, the organoleptic properties of the suggested formulas were evaluated by the formulator himself. These tests were approved by the local ethics committee (Institutional Review Board [IRB] of An-Najah National University). After that, the formula that showed the

**Table 1. Formulation of SiP oral solution.**

| Ingredients | T 1 | T 2 | T 3 | T 4 |
|---|---|---|---|---|
| Sitagliptin Phosphate Monohydrate | 6.4 g | 6.4 g | 6.4 g | 6.4 g |
| Liquid Sorbitol 70% w/v | 140 ml | 140 ml | 140 ml | 140 ml |
| Mannitol | 14 g | 14 g | 14 g | 14 g |
| Peppermint | 2 ml | 0 | 0 | 0 |
| Cherry | 2 g | 3 g | 3 g | 0 |
| Strawberry | 0 | 0 | 2 g | 4 g |
| Purified water up to 640 ml to produce 1% w/v concentration | 640 ml | ml | 640 ml | 640 ml |

most suitable taste, odor, and color was selected for stability studies according to the ICH guidelines for stability studies except analysis time intervals [16].

The SiP solution (T4) was prepared to a final concentration of 10 mg/ml. Three samples were taken from the stored SIP solution for initial analysis and the remaining samples were stored at room temperature and in the stability chamber for stress conditions to analysis at 0, 7, 14, 21, and 28 days.

The T4 oral solution was made in triplicate using 6.4 g of the drug dissolved in 30 ml of distilled water. After that, 140 ml of 70% sorbitol w/v was applied to the previous mixture. Meanwhile, mannitol was continuously mixed into the mixture using a magnetic stirrer. The flavor (Strawberry) was then dissolved in 10 ml of water and added to the resulting mixture. Finally, filtered water was applied until the required amount was achieved, and the mixture was thoroughly mixed to produce a homogenized solution. The obtained solutions were poured into plastic bottles with a capacity of 100 ml. Three bottles were held at room temperature for stability testing, while the other three were put in an accelerated stability chamber (40˚C, 75% RH).

## Quality control of the oral solution

The obtained oral solutions were first examined for clarification, flavor, odor, and color. Chemical and physical stability, assay, pH, and microbial contamination were all studied further using the formula that provided the best results.

## Chemical analysis

According to the validation portion, the amount of SiP in the obtained solution was investigated using a new validated HPLC analytical process. The HPLC experimental conditions were optimized on the Octadecyl silane C18 chemically bonded column (C18 150 X 4.6 porous silica column with 5 μ particle size or equivalent) that was purchased from ACE, (United Kingdom). The mobile phase was prepared by mixing 500 ml of 0.1M potassium dihydrogen phosphate buffer (13.6 g of potassium dihydrogen phosphate dissolved in 1000 ml distilled water) and 500 ml of methanol to make a mobile phase ratio of 1:1. Phosphoric acid was used to adjust the pH to 4.5, after that the solution was degassed and filtered through a 0.45 membrane filter. After being analyzed using a UV-spectrophotometer in the UV range (220–320), the maximum absorption was located at a wavelength of 260 nm (260 nm). The flow rate used was 0.5 ml/minute and the injection volume was 10 μl.

## Analytical validation

The method was validated under the ICH guidelines [17, 18]. Parameters such as system suitability, selectivity, linearity, range, accuracy (recovery), and precision (repeatability) were all validated. For example, separated normal and placebo solutions were analyzed under the same conditions to determine the method's specificity. The results were compared with that of a sample solution that had been analyzed on the same day to check if there is any interference between the SiP peak and any other peak in the chromatogram. To determine the accuracy of the method, three different concentrations were prepared at 60%, 100%, and 140% of the final assay concentration for SiP. Three samples were taken from each concentration. Two injections were given to each sample. For SiP, standard solutions were prepared with the same concentrations as the final assay concentration. The accuracy parameter (percent of recovery) was calculated. On the other hand, the precision parameter was determined on six sample- solutions with a final concentration of 0.1285 mg/ml for SiP. The sample and the standard solutions were prepared as directed in the specificity test. The precision parameter (coefficient of

variation) was calculated. To assess the linearity of the method, five concentrations of the sample solutions were employed for constructing calibration curves covering a concentration ranging from 0.0771 to 0.1799 mg /ml. Concerning ruggedness and robustness, the same trial was used to prepare the sample solutions that were used in each part of the ruggedness and robustness parameter. Standard and sample solutions were prepared and analyzed by two different analysts, different flow rates, different % buffer in the mobile phase, different pH values for the mobile phase, different two instruments, different two columns, and filter versus centrifuge. Parameters of system suitability were determined by injecting six samples each one composed of 10.0 µl of the standard solution with a final concentration of 0.1285 mg/ml SiP. According to the obtained chromatograms, parameters such as injection precision, tailing factor and theoretical plates were calculated. This analytical procedure validation was conducted using three trials of the product and one trial of the placebo product that was prepared in the laboratory.

## Preparation of stock, sample, and standard solutions

**Diluent.** Highly purified water was prepared by using a Millipore Milli-Q Plus water purification system.

**Standard stock solution.** Transfer 64.25 mg of Sitagliptin phosphate monohydrate RS/WS (accurately weighed) to 50 ml volumetric flask, add 40 ml of diluent, stir and sonicate to dissolve, dilute to volume using the same diluent, and mix.

**Standard solution.** Using a volumetric pipette, transfer 5 ml of standard stock solution to 50 ml volumetric flask, diluted to 50 ml using diluent, and mix well.

**Sample solution.** Using a volumetric pipette, transfer 5 ml of the reconstituted solution (64.25 mg of Sitagliptin phosphate monohydrate) to a 100 ml volumetric flask, add 75 ml of diluent, shake and sonicate for 30 min, leave to return to room temperature, dilute with diluent to the volume, and mix. After that, transfer 10 ml of the supernatant to a 50-ml volumetric flask, dilute with diluent to volume, and mix. Pass a portion of this solution through a nylon filter having a 0.45-µm, reject the first 10 ml, and use the filtrate as the Assay preparation.

*Procedure*. Inject 10 µl of the standard solution and sample solution directly into the HPLC column under the optimized chromatographic conditions. Suitability requirements were fixed for column efficiency to be not less than 800 theoretical plates from the SiP peak, tailing factor not more than 2.0, and relative standard deviation below 2.0% for the standard solution.

## Microbial contamination testing

This test was conducted according to the pharmacopeial specification and guidelines [19]. Precisely, to prepare the culture media, 28 g of nutrient agar dehydrated powder was dissolved using 1000 ml of distilled water. The obtained solution was brought to boiling and kept under vigorous mixing. Then, the solution was sterilized by placing it in the autoclave for 15 min at 125˚C. The sterilized solution was then poured into sterilized Petri dishes. Then, the Petri dishes were placed for 24 h in the refrigerator. After that, 0.1 ml of each syrup was placed on the petri dish and incubated for 48 h at 37˚C, and then by visual inspection using the colony counter we detected the presence of *Escherichia coli*, total aerobic microbial count, or total combined yeast and mold counts.

## Results

SiP powder was used to make a T4 oral solution. The pH, assay, related substances, organoleptic properties (Table 2), and microbial contamination of this oral solution were all tested (Table 3). Also, T4 was subjected to further stability studies as reported in Table 2. This

**Table 2. Quality control tests for solutions from pure powder.**

| Q.C. | 0 Day | | 7 Days | | 14 Days | | 21 Days | | 28 Days | |
|---|---|---|---|---|---|---|---|---|---|---|
| | 25˚C | 40˚C | 25 ˚C | 40 ˚C | 25 ˚C | 40 ˚C | 25 ˚C | 40 ˚C | 25 ˚C | 40 ˚C |
| Assay (%w/v) | 98.5 | N/A | 100.2 | 97.47 | 100.7 | 98.8 | 101.7 | 99.8 | 102.1 | 100.2 |
| | ±0.3 | | ±0.36 | ±0.3 | ±0.3 | ±0.4 | ±0.3 | ±0.3 | ±0.3 | ±0.3 |
| **Related substances** (%w/v): Unspecified impurities | | | | | | | | | | |
| | 0.04 | 0.04 | 0.05 | 0.07 | 0.07 | 0.07 | 0.07 | 0.08 | 0.07 | 0.08 |
| Total impurities | 0.04 | 0.04 | 0.05 | 0.07 | 0.07 | 0.07 | 0.07 | 0.08 | 0.07 | 0.08 |
| pH | 3 | 3.1 | 3.1 | 3.5 | 3.2 | 3.6 | 3.3 | 3.7 | 3.5 | 4 |
| Organoleptic Properties | Complies | | Complies | | Complies | | Complies | | Complies | |

formula showed good chemical, physical, and microbial stability during the whole period of study (4 weeks). Indeed, the drug concentration remained unchanged despite being stored under room or accelerated conditions for one month. In addition, all other stability parameters remained stable as per day 0.

Regarding the microbial stability of SiP, no signs of contamination were observed as summarized in Table 3.

Stress degradation studies showed that alkaline degradation is the major degradation pathway of SiP. Actually, the forced degradation analysis aimed to see how well the analytical method could classify and differentiate degradation products from unbroken SiP. The results showed that the HPLC approach used can distinguish the main degradation product (fumarate adduct) with a relative retention time of 1.25 min (Fig 3).

Since all validation parameters were within the acceptable ranges as stated, the used analytical method was found to be accurate for the assay of SiP in the prepared solutions (Table 4). Moreover, the parameters of system suitability were assessed and they were within the acceptance criteria as summarized in Table 5. The related substances test was carried out according to the European Pharmacopeia (EP 9.0) method of analysis [20]. The approach was precise since isolated normal and placebo solutions were tested under the same conditions and the results were compared to those of a reference solution analyzed the same day, with no interference between the SiP peak and any other peak in the chromatogram. The retention time obtained for SiP was about 5. 48 min as reported in Fig 2. Additionally, stress testing was performed under different stress conditions, including degradation by 0.1N HCl, 0.1N NaOH, $H_2O_2$, light, and heat. At all the conditions, there was no interference between the SiP and any other degrades and the purity of the SiP was 0.999 as reported in Table 6. The method also was accurate since the analyzed three different concentrations (60%, 100%, and 140%) of SiP solutions showed the percent of recovery of 99.43%, 101.29%, and 101.02%, respectively. On the other hand, the calculated precision parameter showed a coefficient of variation equal to 0.48%. To determine the method's linearity, a linear calibration curve was created using five SiP concentrations ranging from 0.0771 to 0.1799 mg/ml. The obtained calibration curve showed a correlation coefficient close to 1 and a Y-intercept equal to 11904.150 and a limit of detection equal to 77.1 μg/ml, which ensures that the method is linear within this range.

**Table 3. Microbial limit tests after 28 days of storage at room temperature.**

| Microbiology | Results | Limits |
|---|---|---|
| *Escherichia coli* bacteria | Negative/ml | Absent |
| Total combined yeast and molds count | < 10 cfu/ml | not more than $10^1$ cfu/ml |
| Total aerobic microbial count | < 10 cfu/ml | not more than $10^2$ cfu/ml |

**Table 4. Summary results of the validation of the assay of SiP solution.**

| Parameter | Statistical Measure | Result | Limits |
|---|---|---|---|
| Specificity | No interference between SiP chromatogram and other chromatograms | Complies | No interference between the active material peak and any other peaks. |
| Accuracy | For 60% | 99.43 | 98.00–102.00 |
| | For 100% | 101.29 | |
| | For 140% | 101.02 | |
| Linearity | • Correlation Coefficient | 0.99930 | • Min 0.9950 |
| | • Y-Intercept | 11904.150 | • ±2.0% of the average area |
| Precision | Coefficient of Variation (RSD) | 0.48% | Maximum of 2.0% |

Standard and sample solutions were prepared and evaluated by two separate analysts, with different flow rates, percent buffer in the mobile phase, pH values for the mobile phase, different two instruments, different two columns, and filter versus centrifuge, and no major variations were identified. No significant differences were observed due to these changes as the coefficient of variance for all of them was below the recommended limit of 2% (Tables 4 and 7). The system suitability was determined by injecting 10.0 μl of the standard solution and finding the results within the recommended range. The relative standard deviation on six replicate injections was found to be 0.63%, tailing factor 1.9±0.2%, and the column efficiency expressed by several theoretical plates for the six replicate injections and was 4222.

Once the appropriate formula was assessed for solubility and organoleptic properties, it was stored at two storage conditions, room temperature (25±3˚C) and accelerated conditions (40˚C/75% RH), to assess its stability. Throughout the stability analysis, the solution exhibited chemical, physical, and microbial stability (4 weeks). In fact, there was no appreciable change in pH which remained within the accepted range (3–4) till the end of the study period and the mean % of remained SiP was close to 97% ± 0.2 at room temperature. Also, until the end of the fourth week of storage, bottles stored under stability conditions of 40˚C/75% RH displayed similar stability. Furthermore, there were no visible signs of organoleptic changes such as odor, color, or crystal formation in the solutions. Furthermore, the solutions showed no signs of microbial contamination and no signs of microbial development, as claimed in Table 3.

## Discussion

The pharmacist faces difficulties in community and hospital pharmacy practice with the preparation of liquid dosage forms that are not commercially available for pediatric and geriatric patients who have trouble swallowing tablets or capsules, as well as patients who may obtain medicines through nasogastric or gastrostomy tubes [21]. To the best of our knowledge, SiP is not available in any liquid oral dosage forms that could not be suitable for patients with swallowing difficulties. In fact, during the Covid-19 epidemic where geriatric patients with diabetes are considered highly vulnerable to ending up in the ICU, and having a suitable liquid dosage form for this important drug may be of great help to the physicians. At the beginning of the

**Table 5. Summary results of system suitability.**

| Parameter | Results | Limit |
|---|---|---|
| Injection Precision (RSD) | 0.63 | < 2.0% (for six injections). |
| Tailing Factor | 1.2 | < 2.0 |
| Number of Theoretical Plates | 4222 | > 2000 |
| Capacity Factor | 7.5 | > 2 |

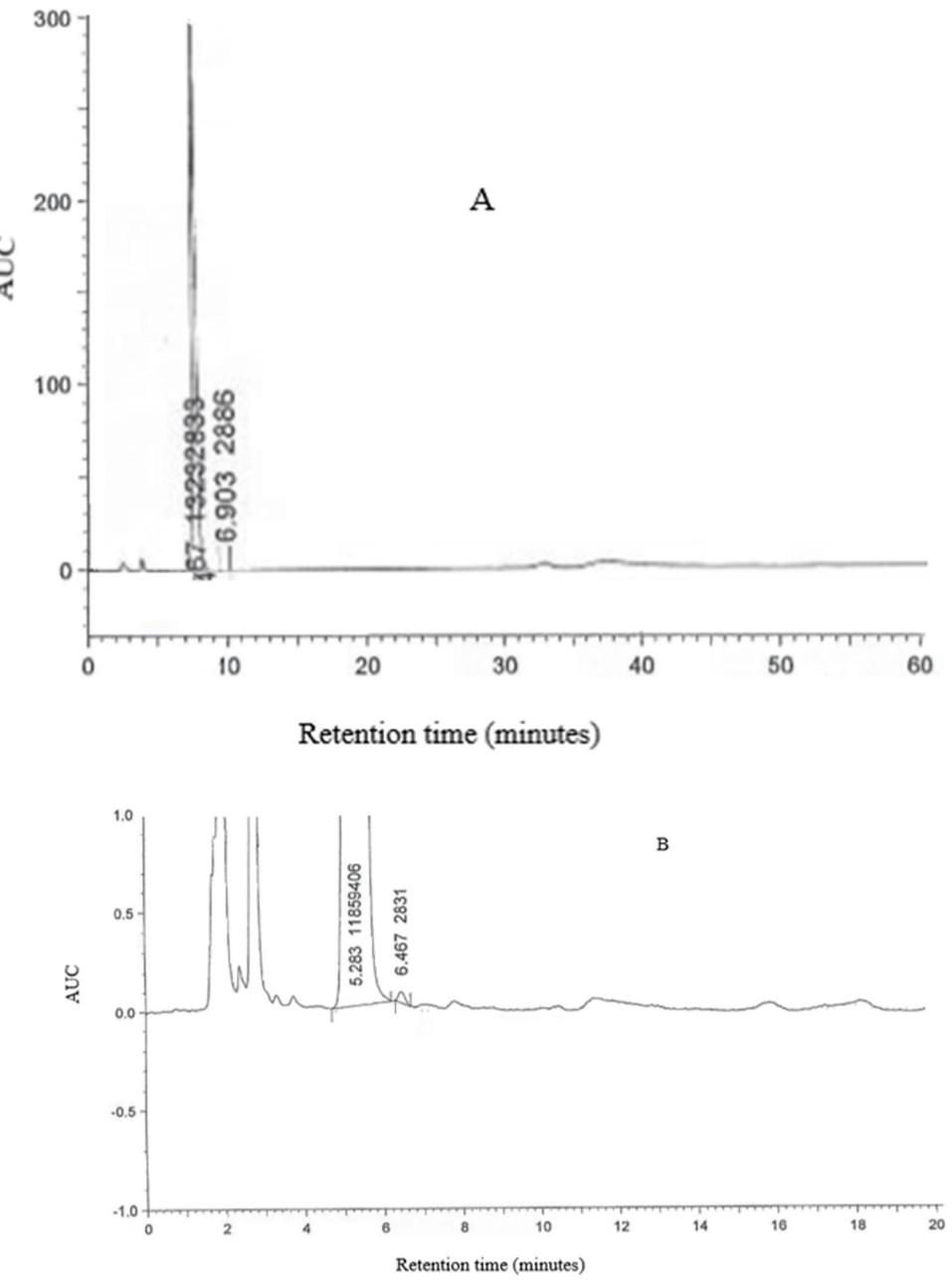

**Fig 2.** HPLC chromatograms of SiP sample (A) and reference standard (B).

development of the oral liquid formulation, it was noted that the SiP bulk powder was freely soluble in water. As a result, an oral solution of this drug can be conveniently prepared without compromising absorption, as is the case for water-insoluble drugs. Furthermore, SiP has an oral bioavailability of about 87%, and its pharmacokinetic parameters are unaffected by food intake. Besides, it reaches maximum plasma concentration within 3 h [22]. Accordingly, there is no need for either dissolution or bioequivalence studies (as in capsules, tablets, and suspensions) to demonstrate that the administered dose is bioequivalent to the brand tablets. On the other hand, a strong bitter taste of the solution was detected. As a result, several attempts were

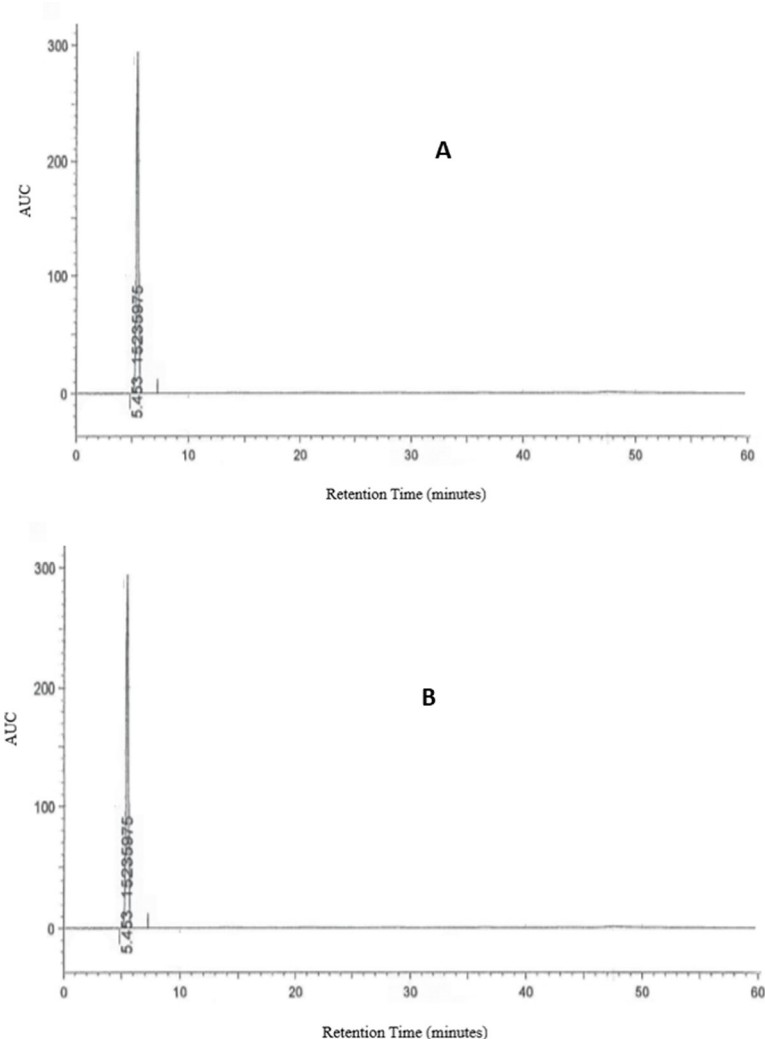

**Fig 3.** HPLC Chromatogram of SiP and its degradation products substance (A and B before and after magnification.

**Table 6. Summary results of SiP stressed conditions.**

| Stress solution | Assay after one day | Degradation of the main peak | Interference with the main peak | Purity check for the main peak | UV spectrum for the main peak |
|---|---|---|---|---|---|
| $H_2O$ kept at 60 ˚C | 99.6% | No | No | 0.999 | Similar |
| 0.1 N NaOH | 86.2% | Partial | No | 0.999 | Similar |
| 3% $H_2O_2$ | 97.6% | No | No | 0.999 | Similar |
| 0.1 N HCl | 96.7% | No | No | 0.999 | Similar |
| At 120˚C | 97.8% | No | No | 0.999 | Similar |
| In UV Chamber | 99.1% | No | No | 0.999 | Similar |

Initial result for the stock solution was = 99.3%.

Retention time for the main peak was = 5.48 min.

**Table 7. Summary ruggedness and robustness results of the assay validation of SiP solution.**

| Ruggedness and Robustness | Coefficient of Variation. (RSD) | Different analyst | Different time analysis | Different Flow rate | Different % buffer in the mobile phase | Different pH for mobile phase | Different instruments | Different Columns | Filter vs centrifuge | Max 2.0% |
|---|---|---|---|---|---|---|---|---|---|---|
| | Coefficient of Variation. (RSD) | 0.84 | 0.10 | 0.41 | 0.10 | 0.54 | 0.53 | 0.61 | 0.69 | |

made to disguise or mitigate this unfavorable bitterness. At the end of our attempts, we successfully developed a new extemporaneous SiP oral liquid formula with suitable organoleptic properties. Also, a new analytical method was developed and validated to assess the quality and stability of the obtained SiP extemporaneous solution using pure SiP powder [23]. Based on validation parameters such as specificity, device linearity, suitability, accuracy, range, precision, and ruggedness, the current validation study aims to ensure that the established SiP solution assay method is successful and reproducible. The validated HPLC method showed that SiP was eluted after 5.45 min of the run while the main related substance was eluted 6.467 (Fig 2A and 2B). A freshly prepared solution of SiP was tested using this HPLC method and correlated with the SiP reference standard. In fact, both retention time and the shape of the peaks were comparable between the reference standard solutions and the sample (Fig 2).

The stress degradation studies showed that alkaline degradation is the major degradation pathway of SiP (Fig 3) [24, 25]. As a result, it's critical to build an HPLC system that can identify degradation products. As a consequence, the approach can be used to investigate the stability of SiP oral solution. After that, the selected formula was assessed for its assay, pH, microbial contamination, and organoleptic properties including color, smell, taste, and turbidity during the whole period of the study (one month) to estimate the expiry date of the obtained formula (Tables 2 and 3). At the end of the 28 days, more than 97% of the initial concentration of SiP was detected in the solutions which were stored in a selected stability condition. Moreover, no observable changes in color, odor or apparent microbial growth were observed in any sample during the stability study's period. Besides, no apparent change in the mean pH was observed in any of the tested samples. Since the SiP liquid dosage form is not commercially available, community and hospital pharmacists may benefit from this knowledge because they can prepare such a formula on a prescription using pure SiP powder. This could be very useful for a wide range of patients who need oral hypoglycemic drugs in a liquid dosage form.

## Conclusion

Using pure SiP powder, an extemporaneous SiP solution was successfully prepared. In terms of stability and organoleptic properties, the used formula was of high quality. The community pharmacist can extemporaneously prepare the solution using this active ingredient from the sponsor of the generic. Scientific details and instructions on how to compound commercial immediate-release tablets into oral solution, as well as information on the solution's expiration date, should be provided in pharmaceutical manufacturers' product inserts.

## Acknowledgments

The authors would like to acknowledge An-Najah National University for its support.

## Author Contributions

**Conceptualization:** Abdel Naser Zaid.

**Data curation:** Nidal Jaradat.

**Investigation:** Tharaa Abu Salah, Sameera Siaj, Lana Omari.

**Methodology:** Yara Abu Zaaror.

**Project administration:** Abdel Naser Zaid.

**Supervision:** Abdel Naser Zaid, Yara Abu Zaaror.

**Validation:** Aiman Kaddumi, Mashhour Ghanem.

**Writing – original draft:** Tharaa Abu Salah, Sameera Siaj, Lana Omari.

**Writing – review & editing:** Abdel Naser Zaid, Nidal Jaradat.

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
