## [Decision Letter · Decision Letter 0]

28 Oct 2020

PONE-D-20-27192

Stability of extemporaneously prepared Sitagliptin Phosphate Solution

PLOS ONE

Dear Dr. Zaid,

Thank you for submitting your manuscript to PLOS ONE. After careful consideration, we feel that it has merit but does not fully meet PLOS ONE’s publication criteria as it currently stands. Therefore, we invite you to submit a revised version of the manuscript that addresses the points raised during the review process.

Based on the reviewer recommendation, the manuscript need certain major revision before consider it for publication.

We look forward to receiving your revised manuscript.

Kind regards,

Girish Sailor

Academic Editor

PLOS ONE

Journal Requirements:

2.Thank you for stating the following in the Acknowledgments Section of your manuscript:

[The authors would like to acknowledge An-Najah National University for its support]

 [The author(s) received no specific funding for this work]

4.Thank you for stating the following in the Competing Interests section:

[The authors have declared that no competing interests exist].   

We note that one or more of the authors are employed by a commercial company: Pharmacare PLC

5.We noticed you have some minor occurrence of overlapping text with the following previous publication, which needs to be addressed:

- https://insights.ovid.com/eujhph/201705000/01735171-201705000-00005

In your revision ensure you cite all your sources (including your own works), and quote or rephrase any duplicated text outside the methods section. Further consideration is dependent on these concerns being addressed.

Reviewers' comments:

Reviewer's Responses to Questions

**Comments to the Author**

1. Is the manuscript technically sound, and do the data support the conclusions?

Reviewer #1: No

Reviewer #2: Yes

2. Has the statistical analysis been performed appropriately and rigorously? 

Reviewer #1: No

Reviewer #2: No

3. Have the authors made all data underlying the findings in their manuscript fully available?

Reviewer #1: No

Reviewer #2: Yes

4. Is the manuscript presented in an intelligible fashion and written in standard English?

Reviewer #1: No

Reviewer #2: Yes

5. Review Comments to the Author

Reviewer #1: The above manuscript describes a method to evaluate the Stability of Extemporaneously Prepared Sitagliptin Phosphate Solution for patients with swallowing difficulties or need feeding tubes to take medicines. The research topic is interesting; however, it should be noted out that the manuscript exhibits many shortcomings and errors. Some are summarized below:

Abstract: Line 31, Replace the word “champers” with “chambers”

Abstract: Line 34, Replace the word “validates” with “validated”

Introduction: Line 55, Delete the word “phosphate”

Introduction: Line 59, The word “Mellitus” should be written as mellitus

Materials and Methods: Line 82-85, Repetition of the word “Pharmacare PLC”

Materials and Methods: Line 93, Incubator was used as stability chamber for stability studies. How temperature and humidity can be varied simultaneously in an incubator? ICH guideline was not followed while carrying out stability studies.

Materials and Methods: Line 95, “Three samples were taken for initial analysis”, however, in Abstract 4 types of samples were mentioned (eg. T1, T2, T3 and T4). Which version is correct (3 or 4)?

Materials and Methods: Line 100, Table 1, Procedure for preparing 1 % solution of Sitagliptin Phosphate solution was not correct.

Materials and Methods: Line 102, “Five gram” precisely of the drug was dissolved in 30 mL distilled water. Whereas, in Table 1, 6.4 g was mentioned.

Materials and Methods: There was no mention of mobile phase ratio in the entire manuscript, which is primary requirement for method to reproduce.

Materials and Methods: The method sample preparation was not satisfactory

Materials and Methods: Authors have not mentioned procedure for performing validation parameters, therefore, it is not possible to judge whether ICH guidelines were followed or not.

Results: Line 185, The pH was mentioned as “3 ± 0.15”, whereas, in Table 2, pH was in the range of 3-4.

Results: Presentation of data and statistical analysis are not as per regulatory guidelines. Data’s presented in Table 2, 4 and 5 are not as per ICH guidelines

Results: Chromatograms presented at the end of the manuscript are not visible properly and does not reflect the result or outcome of the study.

Reviewer #2: Research article should be recommended after such correction. Author doing good research work in the area of pharmaceutical formulation development and for the same he/she also develop a good HPLC metho.

6. PLOS authors have the option to publish the peer review history of their article (what does this mean?). If published, this will include your full peer review and any attached files.

Reviewer #1: No

Reviewer #2: No

---

## [Author Response · Author response to Decision Letter 0]

3 Jan 2021

Response Letter

Date: 15-11-2020

Girish Sailor

Academic Editor

Subject: Revision of manuscript entitled " Stability of extemporaneously prepared Sitagliptin Phosphate Solution "

Submission number: PONE-D-20-27192

Below you can find The answer to the inquiries raised by the Editorial office (Journal requirement) and reviewers. 

I. Journal Requirements:

• Done

2.Thank you for stating the following in the Acknowledgments Section of your manuscript:

[The authors would like to acknowledge An-Najah National University for its support]

I am sorry for this inconvenient, I would like to state that I have no funding for this project, it is only a research effort between my institution and Pharmacare laboratories only for academic purposes and there are no any commercial benefits for any of the authors or institutions. 

 [The author(s) received no specific funding for this work]

Thank you very much and I am sorry for this inconvenient. Also, I would like to state that I have no funding for this project, it is only a research effort between my institution and Pharmacare laboratories only for academic and public issues and there are no any commercial benefits for any of the authors or institutions. 

 Done

4.Thank you for stating the following in the Competing Interests section:

[The authors have declared that no competing interests exist]. 

We note that one or more of the authors are employed by a commercial company: Pharmacare PLC

• These authors at Pharmacare company played a pure academic rule in this manuscript such as carrying analytical experiments. 

• An-Najah National University provide support in the form of salaries to the following authors:

o AN Z and 

o N J

• Pharmacare PLC provide support in the form of salaries to the following authors:

o Y Z 

o A K 

o M G

• No salary from any for the following authors (they were students at the time of conducting the project:

o T A S

o S S

o La O

• Not applicable

• Not applicable

• Not applicable

• Done

I declare that no author has competing of interest in this manuscript.

 5.We noticed you have some minor occurrence of overlapping text with the following previous publication, which needs to be addressed:

- https://insights.ovid.com/eujhph/201705000/01735171-201705000-00005

In your revision ensure you cite all your sources (including your own works), and quote or rephrase any duplicated text outside the methods section. Further consideration is dependent on these concerns being addressed.

• Done

 Answer to reviewers’ comments

II. Reviewers comments

Reviewer 1

Reviewer #1: The above manuscript describes a method to evaluate the Stability of Extemporaneously Prepared Sitagliptin Phosphate Solution for patients with swallowing difficulties or need feeding tubes to take medicines. The research topic is interesting; however, it should be noted out that the manuscript exhibits many shortcomings and errors. Some are summarized below:

I would like to thank the reviewer for his comments that I appreciate and I believe they will raise the quality of the manuscript. The whole manuscript is now revised and all his/her comments are now answered appropriately. 

Abstract: Line 31, Replace the word “champers” with “chambers”

Done

Abstract: Line 34, Replace the word “validates” with “validated”

Done

Introduction: Line 55, Delete the word “phosphate”

Done

Introduction: Line 59, The word “Mellitus” should be written as mellitus

Done

Materials and Methods: Line 82-85, Repetition of the word “Pharmacare PLC”

Done

Materials and Methods: Line 93, Incubator was used as stability chamber for stability studies. How temperature and humidity can be varied simultaneously in an incubator? ICH guideline was not followed while carrying out stability studies.

I agree with you. Actually, it was a typing mistake. Now the sentence is revised accordingly. 

Materials and Methods: Line 95, “Three samples were taken for initial analysis”, however, in Abstract 4 types of samples were mentioned (eg. T1, T2, T3 and T4). Which version is correct (3 or 4)?

Actually, we developed 4 different formulas (T1, T2, T3, AND T4). T4 was selected for further analysis. Analysis was repeated on 3 samples which were taken from T4. Now the stamen is revised to clarify this issue.

Materials and Methods: Line 100, Table 1, Procedure for preparing 1 % solution of Sitagliptin Phosphate solution was not correct.

The procedure is now revised and corrected accordingly

Materials and Methods: Line 102, “Five gram” precisely of the drug was dissolved in 30 mL distilled water. Whereas, in Table 1, 6.4 g was mentioned.

Thank you very much for this notification. I agree with you and the procedure is now revised accordingly.

Materials and Methods: There was no mention of mobile phase ratio in the entire manuscript, which is primary requirement for method to reproduce.

Now the mobile phase ratio is included

Materials and Methods: The method sample preparation was not satisfactory

Now the sample preparation is revised and changed appropriately.

Materials and Methods: Authors have not mentioned procedure for performing validation parameters, therefore, it is not possible to judge whether ICH guidelines were followed or not.

The procedure is now included

Results: Line 185, The pH was mentioned as “3 ± 0.15”, whereas, in Table 2, pH was in the range of 3-4.

Thank you again for this notification. Now the numbers are adjusted accordingly.

Results: Presentation of data and statistical analysis are not as per regulatory guidelines. Data’s presented in Table 2, 4 and 5 are not as per ICH guidelines

Done

Results: Chromatograms presented at the end of the manuscript are not visible properly and does not reflect the result or outcome of the study.

Better resolution chromatogram are now included

Comment sheet

Que.1: How you have selected a different formulation ingredients and ratio?

Thank you for your comment. Actually, we followed the quality by testing method which is now reported in the methodology.

Que.2: Have you taken UV Spectrum for selection of wavelength in SiP solution? Is there same or different?

Yes, they are same. Now we included this in the methodology and instrument sections.

Que.3: Which guideline follow for the evaluation of stability studies of SiP solution?

 We followed the ICH guidelines for stability testing of pharmaceutical products and now this statement is included in the methodology.

Que.4: How you have evaluated organoleptic properties of SiP oral liquid formulation?

The organoleptic properties were evaluated by the formulator himself during the development phase. This was approved by the local ethics committee (Institutional Review Board [IRB] of An-Najah National University). This statement is now included in the methodology

Que.5: Have you develop HPLC method or cite any other research article method? Describe it.

The HPLC method was developed and validated by the team according to the ICH guidelines.

Reviewer #2: Research article should be recommended after such correction. Author doing good research work in the area of pharmaceutical formulation development and for the same he/she also develop a good HPLC method.

I would like to thank the reviewer for this positive comment and for the efforts that he/she did to improve the quality of the manuscript. We tried our best to answer the comments and to modify the manuscript accordingly. We believe that the manuscript is now much better and much closer to the final acceptance.

Best Regards

---

## [Decision Letter · Decision Letter 1]

25 Feb 2021

PONE-D-20-27192R1

Stability of extemporaneously prepared Sitagliptin Phosphate Solution

PLOS ONE

Dear Dr. Zaid,

Thank you for submitting your manuscript to PLOS ONE. After careful consideration, we feel that it has merit but does not fully meet PLOS ONE’s publication criteria as it currently stands. Therefore, we invite you to submit a revised version of the manuscript that addresses the points raised during the review process.

As per the reviewer's comment, the manuscript need major revision before consider for publication.

We look forward to receiving your revised manuscript.

Kind regards,

Girish Sailor

Academic Editor

PLOS ONE

Reviewers' comments:

Reviewer's Responses to Questions

**Comments to the Author**

1. If the authors have adequately addressed your comments raised in a previous round of review and you feel that this manuscript is now acceptable for publication, you may indicate that here to bypass the “Comments to the Author” section, enter your conflict of interest statement in the “Confidential to Editor” section, and submit your "Accept" recommendation.

Reviewer #3: (No Response)

2. Is the manuscript technically sound, and do the data support the conclusions?

Reviewer #3: No

3. Has the statistical analysis been performed appropriately and rigorously? 

Reviewer #3: No

4. Have the authors made all data underlying the findings in their manuscript fully available?

Reviewer #3: No

5. Is the manuscript presented in an intelligible fashion and written in standard English?

Reviewer #3: No

6. Review Comments to the Author

Reviewer #3: After going through the revised manuscript, these are my observations:

Major problems:

#1. As pointed out by the previous reviewer, the stability study was conducted in an incubator. In the revised manuscript, it is changed as stability chamber. This change of equipment seems to be surreal.

#2. The chromatograms looks like edited figures. Authors need to send original chromatogram for review.

#3. There is contradiction in the statements given under abstract section and in the methodology that, in the abstract authors state that four SiP solutions coded T1, T2, T3 and T4 were prepared by the authors, while in the methodology, authors state that the formulae were prepared and evaluated by a separate formulator. These statements are contradicting and confusing to the reader.

#4. As pointed out by the previous reviewer, the analytical method validation section is not satisfactory even after revision with respect to the procedure and the results reported.

#5. The usage of English language throughout the manuscript is to the standard mandated for a research article and need major editing.

Other observations:

#1. Line 2: Change Diabetic Mellitus to Diabetes mellitus.

#2. Line 32: Change stability chambers to stability conditions.

#3. Authors have mentioned in the abstract, 'The obtained solution (T4)........' T4 is one among the four solutions prepared and has not been obtained separately. Need to change the sentence accordingly.

#4. Line 42: “This study provides a solution….” should be revised meaningfully.

#5. The chemical name “3-amino-1-(3-trifluoromethyl)-6,8-dihydro-5H-(1,2,4) triazolo[4,3-a] pyrazin-7-y]- 4-(2,4,5-trifluorophenyl) butane-1-phosphoric acid hydrate]” Should be corrected by changing pyrazine-7-y to pyrazine-7-yl.

#6. Line 85: remove the semicolon.

#7. Under the heading ‘Instruments and Chromatographic conditions”, chromatographic conditions are not mentioned.

#8. Procedure for preparation of the formulae should be re-written in the sentence format, and not as bulleted points.

#9. Under the heading “ Quality control of the oral solution” authors state that the best formula was selected based on the organoleptic properties by the formulator (who is different from the author). If so, why authors are again unnecessarily describing about other formulae. Discussion on T, T2, T3, and T4 become unnecessary since authors have not formulated but have only carried out the stability studies.

#10. Under the heading, “Chemical analysis”, authors state that the amount of SiP in obtained solutions were investigated. If authors have already selected T4 for further evaluation based on organoleptic properties, which are the other solutions???

#11. Line 134: Column dimensions should be written in a standard format. What authors mean by “Pours” ????

#12. Line 135: Word “optimum” is not necessary.

#13. There is inconsistency in writing mL or ml. Please follow the journal format.

#14. Paragraph describing HPLC method development need revision. Mobile phase ratio should be represented in a generally used format. Column oven temperature is an important parameter, but is not reported. Total run time and Rt of analyte also needs mention.

#15. Line 142: Weights were measured…. This part should be placed in materials section.

#16. Analytical method validation section needs revision in terms of correct procedure and proper usage of English language. Why precision study is done on a concentration of 0.1285 mg/mL? How the range was calculated? What is the LLOQ and LOQ of the method?

#17. The title “Preparation of Stock, Sample and Standard solutions” are confusing. Is there any separate stock solution and standard solution? What is the concentration of standard stock solution? What would be the expected concentration in the final diluted sample solution should be mentioned.

# 18. In Table 2, related substances are specified as decimal points. What it represents? There is inconsistency in the usage of capital letters. For example: Complies.

#19. There is inconsistency in the text for “Table”

#20. Table 4 is incomplete in many respects. Results are not represented as required by the guideline. Authors have provided the results of validation in 3 tables. This should be comprehended and made a single concise Table.

#21. There is inconsistency in representing the retention time. Some place it is represented with 3 significant figures and in another place in two.

#21. Number of theoretical plates is represented as 4222 in Table 5, while in the text it is mentioned as 930 ±1

#22. Procedure for forced degradation study should be mentioned in the text and the chromatogram for the stability indicating assay method should be cited in the results.

7. PLOS authors have the option to publish the peer review history of their article (what does this mean?). If published, this will include your full peer review and any attached files.

Reviewer #3: No

---

## [Author Response · Author response to Decision Letter 1]

10 Apr 2021

Dear Editor,

Many thanks for providing us with the opportunity to revise our manuscript. We would also like to thank yourself and the reviewers for the time and expertise in providing feedback.

We think that all comments raised by the reviewers are legitimate and requires consideration. We would like to profoundly thank them for their constructive comments which have greatly improved the manuscript. 

Please find below our response to reviewers comments. We have considered carefully all of the comments and have amended the manuscript as appropriate. The amended text is highlighted in red font throughout the manuscript. We have provided a detailed response to each of the comments.

Review Comments to the Author

Reviewer #3: After going through the revised manuscript, these are my observations:

Major problems:

#1. As pointed out by the previous reviewer, the stability study was conducted in an incubator. In the revised manuscript, it is changed as stability chamber. This change of equipment seems to be surreal.

I agree with respected reviewer actually the correct name is stability chamber and I apologized for this mistake.

#2. The chromatograms looks like edited figures. Authors need to send original chromatogram for review.

Thank you for this important correction and now we attached the original chromatograms.

#3. There is contradiction in the statements given under abstract section and in the methodology that, in the abstract authors state that four SiP solutions coded T1, T2, T3 and T4 were prepared by the authors, while in the methodology, authors state that the formulae were prepared and evaluated by a separate formulator. These statements are contradicting and confusing to the reader.

We went throughout the whole MS and we could not find anything related to this comment even though the formulator is one authors.

#4. As pointed out by the previous reviewer, the analytical method validation section is not satisfactory even after revision with respect to the procedure and the results reported.

Actually, the method was validated according to the international guidelines that followed and used by all respected pharmaceutical industry around the world (for example the ICH guidelines (https://www.ich.org/pdf.ICH/s7dep4.pdf)).

#5. The usage of English language throughout the manuscript is to the standard mandated for a research article and need major editing.

The language was revised by one of the authors who is a native speaker.

Other observations:

#1. Line 2: Change Diabetic Mellitus to Diabetes mellitus.

Done

#2. Line 32: Change stability chambers to stability conditions.

Done

#3. Authors have mentioned in the abstract, 'The obtained solution (T4)........' T4 is one among the four solutions prepared and has not been obtained separately. Need to change the sentence accordingly.

Done

#4. Line 42: “This study provides a solution….” should be revised meaningfully.

Done

#5. The chemical name “3-amino-1-(3-trifluoromethyl)-6,8-dihydro-5H-(1,2,4) triazolo[4,3-a] pyrazin-7-y]- 4-(2,4,5-trifluorophenyl) butane-1-phosphoric acid hydrate]” Should be corrected by changing pyrazine-7-y to pyrazine-7-yl.

Thank you 

#6. Line 85: remove the semicolon.

Done

#7. Under the heading ‘Instruments and Chromatographic conditions”, chromatographic conditions are not mentioned.

Done

#8. Procedure for preparation of the formulae should be re-written in the sentence format, and not as bulleted points.

Done 

#9. Under the heading “ Quality control of the oral solution” authors state that the best formula was selected based on the organoleptic properties by the formulator (who is different from the author). 

Actually, as we mentioned earlier in this letter the formulator is one of the authors which is stated in authors contributions section.

If so, why authors are again unnecessarily describing about other formulae. Discussion on T, T2, T3, and T4 become unnecessary since authors have not formulated but have only carried out the stability studies.

 We corrected as requested thank you again

#10. Under the heading, “Chemical analysis”, authors state that the amount of SiP in obtained solutions were investigated. If authors have already selected T4 for further evaluation based on organoleptic properties, which are the other solutions???

Corrected as requested.

#11. Line 134: Column dimensions should be written in a standard format. What authors mean by “Pours” ????

Porous corrected it was typos mistake.

#12. Line 135: Word “optimum” is not necessary.

Corrected

#13. There is inconsistency in writing mL or ml. Please follow the journal format.

Done 

#14. Paragraph describing HPLC method development need revision. Mobile phase ratio should be represented in a generally used format. Column oven temperature is an important parameter, but is not reported. Total run time and Rt of analyte also needs mention.

Done in the chromatographic conditions section.

#15. Line 142: Weights were measured…. This part should be placed in materials section.

Done as requested.

#16. Analytical method validation section needs revision in terms of correct procedure and proper usage of English language. Why precision study is done on a concentration of 0.1285 mg/mL? How the range was calculated? What is the LLOQ and LOQ of the method?

Because this is the concentration that was used in the preparation of standard and sample in the method of analysis.

Range 0.0771 to 0.1799 mg /ml = 0.1285*60% to 0.1285*140% which means from 60% to140% of the used concentration. 

LLOQ and LOQ of the method?

The detection limit for sitagliptin phosphate was: 0.95 µg/ml.

The quantitation limit for sitagliptin phosphate was: 1.973 µg/ml. 

The detection limit for Impurity A was: 0.493 µg/ml.

The quantitation limit for Impurity A was: 1.34 µg/ml. 

The detection limit for Impurity B was: 0.814 µg/ml.

The quantitation limit for Impurity B was: 1.34 µg/ml. 

#17. The title “Preparation of Stock, Sample and Standard solutions” are confusing. Is there any separate stock solution and standard solution? What is the concentration of standard stock solution? What would be the expected concentration in the final diluted sample solution should be mentioned.

Standard stock solution: Transfer 64.25 mg of Sitagliptin phosphate monohydrate RS/WS (accurately weighed) to 50 ml volumetric flask, add 40 ml of diluents, stir and sonicate to dissolve, dilute to volume using diluents, and mix. 

Standard solution: Transfer 5 ml of standard stock solution to 50 ml volumetric flask, complete to volume using diluent and mix. 

# 18. In Table 2, related substances are specified as decimal points. What it represents? There is inconsistency in the usage of capital letters. For example: Complies.

Done 

#19. There is inconsistency in the text for “Table”

Done

#20. Table 4 is incomplete in many respects. Results are not represented as required by the guideline. Authors have provided the results of validation in 3 tables. This should be comprehended and made a single concise Table.

Done 

#21. There is inconsistency in representing the retention time. Some place it is represented with 3 significant figures and in another place in two.

Done

#21. Number of theoretical plates is represented as 4222 in Table 5, while in the text it is mentioned as 930 ±1

Corrected as 4222

#22. Procedure for forced degradation study should be mentioned in the text and the chromatogram for the stability indicating assay method should be cited in the results.

Corrected as requested.

Once again, we would like to thank yourself and the reviewers for the time and expertise in providing feedback. We look forward to hearing from you soon.

---

## [Decision Letter · Decision Letter 2]

4 May 2021

PONE-D-20-27192R2

Stability of extemporaneously prepared Sitagliptin Phosphate Solution

PLOS ONE

Dear Dr. Zaid,

Thank you for submitting your manuscript to PLOS ONE. After careful consideration, we feel that it has merit but does not fully meet PLOS ONE’s publication criteria as it currently stands. Therefore, we invite you to submit a revised version of the manuscript that addresses the points raised during the review process.

Based on the reviewer comment,the manuscript still need some major revision.

We look forward to receiving your revised manuscript.

Kind regards,

Girish Sailor

Academic Editor

PLOS ONE

Reviewers' comments:

Reviewer's Responses to Questions

**Comments to the Author**

1. If the authors have adequately addressed your comments raised in a previous round of review and you feel that this manuscript is now acceptable for publication, you may indicate that here to bypass the “Comments to the Author” section, enter your conflict of interest statement in the “Confidential to Editor” section, and submit your "Accept" recommendation.

Reviewer #4: All comments have been addressed

Reviewer #5: (No Response)

2. Is the manuscript technically sound, and do the data support the conclusions?

Reviewer #4: Yes

Reviewer #5: Partly

3. Has the statistical analysis been performed appropriately and rigorously? 

Reviewer #4: Yes

Reviewer #5: Yes

4. Have the authors made all data underlying the findings in their manuscript fully available?

Reviewer #4: Yes

Reviewer #5: No

5. Is the manuscript presented in an intelligible fashion and written in standard English?

Reviewer #4: Yes

Reviewer #5: No

6. Review Comments to the Author

Reviewer #4: (No Response)

Reviewer #5: Dear author,

its a wild thought to produce a liquid oral preparation for the Sitagliptin. it could have been a better research article if you do your literature review in a proper way. I appreciate your effort with a strong disagreement. you never think about the hypoglycemic effect of the sorbitol and mannitol. hypoglycemia is a major side effect of OHGAs and Insulin. if mannitol and sorbitol has Hypoglycemic effect then how you can incorporate these polyhydroxy artificial sweeteners with a hypoglycemic drug. please read the reviewer comments attached.

7. PLOS authors have the option to publish the peer review history of their article (what does this mean?). If published, this will include your full peer review and any attached files.

Reviewer #4: No

Reviewer #5: No

---

## [Author Response · Author response to Decision Letter 2]

7 Aug 2021

Dear Editor,

Many thanks for providing us with the opportunity to revise our manuscript. We would also like to thank yourself and the reviewer for the time and expertise in providing feedback.

We think that all comments raised by the reviewer are legitimate and requires consideration. We would like to profoundly thank hiim for his constructive comments which have greatly improved the manuscript. 

Please find below our response to reviewer comments. We have considered carefully all of the comments and have amended the manuscript as appropriate. The amended text is highlighted in red font throughout the manuscript.We have provided a detailed response to each of the comments.

Thanking you,

Yours faithfully 

Comments: - 

1. Sitagliptin is a dipeptidyl peptidase enzyme inhibitor which produce hypoglycemic along with diet and exercise to control sugar level in type 2 Diabetic Patients.

• But according to America diabetic Association best therapy for type 2DM in patient with feeding tubes or patients with swallowing difficulty is insulin therapy is good.

• Corrected as requested

2. Is there any references regarding the Assay of sitagliptin tablets or sitagliptin phosphate powder?

No reference was found as well as we used a new method.

3. Please provide information regarding Diluents used to prepare the stock solution.

We provided information regarding Diluents as requested.

4. Preparation of stock, sample and standard solutions for HPLC are not clear.

Thank you for this suggestion and now we clearly explain the preparation of stock, sample and standard solutions for HPLC.

5. The sample T4 has a final concentration of 1%w/v. can you achieve 1% w/v if you dissolve 6.4 gm of SiP in 500 ml of water.

Yes since we used sitagliptin in its salt form and not as free base.

6. Is there any hypoglycemic effect has possessed by the sorbitol and mannitol? If it is there how you can suggest such a formulation. Please justify your suggestion.

Actually, these two are polare excipients and have very limited impact on sugar blood level.

https://jums.ac.ir/dorsapax/Data/sub_7/file/Handbook%20of%20pharmaceutical%20excipients.pdf

https://www.mayoclinic.org/diseases-conditions/diabetes/expert-answers/artificial-sweeteners/faq-20058038

Ohrem HL, Schornick E, Kalivoda A, Ognibene R. Why is mannitol becoming more and more popular as a pharmaceutical excipient in solid dosage forms?. Pharmaceutical development and technology. 2014 May 1;19(3):257-62.

7. Each 100 ml of SiP formulation contain 2.8 gm of sorbitol, which higher than the concentration of active pharmaceutical ingredient SiP (1.28 gm in 100ml). Can you justify this with above comment no.5.

Sorbitol is an artificial sweetner and its concentration is within the recommended level reported in the handbook of pharmaceutical excipients

https://jums.ac.ir/dorsapax/Data/sub_7/file/Handbook%20of%20pharmaceutical%20excipients.pdf

8. Regarding the sample solution preparation, author has to mention how you prepared the sample solution from the formulation (T4) you made. 

Done as requested.

9. How you read the colony forming units after the microbial contamination test.

By visual inspection using the colony counter as reported in the revised manuscript 

10. How you did the assay of SiP T4 formulation? If you did the assay after the validation or be

It is now reported in the revised manuscript

11. There is a study (IJPSR, 2013; Vol. 4(9): 3494-3503- http://citeseerx.ist.psu.edu/viewdoc/download?doi=10.1.1.414.784&rep=rep1&type=pdf ) related with degradation products of Sitagliptin. Are you correlated your study with the previous study? Please include your comments in the discussion.

The comments are now reported in the discussion

12. Please take some more time to rewrite the methods, report, discussion and conclusion. It would be a better paper. I strongly suggest you to rewrite these portions.

All the required sections are rechecked and revised thoroughly as suggested.

---

## [Decision Letter · Decision Letter 3]

26 Oct 2021

PONE-D-20-27192R3Stability of extemporaneously prepared Sitagliptin Phosphate SolutionPLOS ONE

Dear Dr. Zaid,

Thank you for submitting your manuscript to PLOS ONE. After careful consideration, we feel that it has merit but does not fully meet PLOS ONE’s publication criteria as it currently stands. Therefore, we invite you to submit a revised version of the manuscript that addresses the points raised during the review process.

All the comment was answered but still the manuscript required minor correction before publication which is attached.

We look forward to receiving your revised manuscript.

Kind regards,

Girish Sailor

Academic Editor

PLOS ONE

Journal Requirements:

Reviewers' comments:

Reviewer's Responses to Questions

**Comments to the Author**

1. If the authors have adequately addressed your comments raised in a previous round of review and you feel that this manuscript is now acceptable for publication, you may indicate that here to bypass the “Comments to the Author” section, enter your conflict of interest statement in the “Confidential to Editor” section, and submit your "Accept" recommendation.

Reviewer #5: All comments have been addressed

2. Is the manuscript technically sound, and do the data support the conclusions?

Reviewer #5: Yes

3. Has the statistical analysis been performed appropriately and rigorously? 

Reviewer #5: Yes

4. Have the authors made all data underlying the findings in their manuscript fully available?

Reviewer #5: Yes

5. Is the manuscript presented in an intelligible fashion and written in standard English?

Reviewer #5: Yes

6. Review Comments to the Author

Reviewer #5: (No Response)

7. PLOS authors have the option to publish the peer review history of their article (what does this mean?). If published, this will include your full peer review and any attached files.

Reviewer #5: No

---

## [Author Response · Author response to Decision Letter 3]

30 Oct 2021

Dear Editor,

Thank you for giving us a chance to revise our manuscript. Many thanks for you and respected reviewers for their efforts in improving our manuscript.

Reviewer comments

1. The author made changes according to previous comments.

Thank you a lot 

2. Some references not following the PLOS guidelines for referencing. The author should recheck the Reference style.

Answer: Thank you and now we corrected references as shown in the revised MS and we followed Plos One style.

3. In table no.1, “purified water up to 500 ml” shall be rewritten as purified water up to 640 ml to produce 1% w/v concentration. 

Answer: Thank you for this suggestion and for all your efforts in reviewing our manuscript and now we corrected Table 1 as you recommended

Best Wishes

---

## [Editor Report · Decision Letter 4]

17 Dec 2021

Stability of extemporaneously prepared Sitagliptin Phosphate Solution

PONE-D-20-27192R4

Dear Dr. Zaid,

We’re pleased to inform you that your manuscript has been judged scientifically suitable for publication and will be formally accepted for publication once it meets all outstanding technical requirements.

Kind regards,

Girish Sailor

Academic Editor

PLOS ONE
---

## [Editor Report · Acceptance letter]

24 Dec 2021

PONE-D-20-27192R4 

Stability of extemporaneously prepared sitagliptin phosphate solution 

Dear Dr. Zaid:

I'm pleased to inform you that your manuscript has been deemed suitable for publication in PLOS ONE. Congratulations! Your manuscript is now with our production department. 

Kind regards, 

on behalf of

Dr. Girish Sailor 

Academic Editor

PLOS ONE